# Acute Oral Toxicity of Pinnatoxin G in Mice

**DOI:** 10.3390/toxins12020087

**Published:** 2020-01-28

**Authors:** Silvio Sosa, Marco Pelin, Federica Cavion, Fabienne Hervé, Philipp Hess, Aurelia Tubaro

**Affiliations:** 1Department of Life Sciences, University of Trieste, Via A. Valerio 6, 34127 Trieste, Italy; mpelin@units.it (M.P.); federica.cavion@phd.units.it (F.C.); tubaro@units.it (A.T.); 2Ifremer, Laboratoire Phycotoxines, Centre Atlantique, 44311 Nantes CEDEX, France; fabienne.herve@ifremer.fr (F.H.); philipp.hess@ifremer.fr (P.H.)

**Keywords:** *Vulcanodinium rugosum*, pinnatoxin G, dinoflagellates, harmful algae, oral toxicity

## Abstract

Pinnatoxin G (PnTx-G) is a marine cyclic imine toxin produced by the dinoflagellate *Vulcanodinium rugosum*, frequently detected in edible shellfish from Ingril Lagoon (France). As other pinnatoxins, to date, no human poisonings ascribed to consumption of PnTx-G contaminated seafood have been reported, despite its potent antagonism at nicotinic acetylcholine receptors and its high and fast-acting toxicity after intraperitoneal or oral administration in mice. The hazard characterization of PnTx-G by oral exposure is limited to a single acute toxicity study recording lethality and clinical signs in non-fasted mice treated by gavage or through voluntary food ingestion, which showed differences in PnTx-G toxic potency. Thus, an acute toxicity study was carried out using 3 h-fasted CD-1 female mice, administered by gavage with PnTx-G (8–450 µg kg^−1^). At the dose of 220 µg kg^−1^ and above, the toxin induced a rapid onset of clinical signs (piloerection, prostration, hypothermia, abdominal breathing, paralysis of the hind limbs, and cyanosis), leading to the death of mice within 30 min. Except for moderate mucosal degeneration in the small intestine recorded at doses of 300 µg kg^−1^, the toxin did not induce significant morphological changes in the other main organs and tissues, or alterations in blood chemistry parameters. This acute oral toxicity study allowed to calculate an oral LD_50_ for PnTx-G equal to 208 μg kg^−1^ (95% confidence limits: 155–281 µg kg^−1^) and to estimate a provisional NOEL of 120 µg kg^−1^.

## 1. Introduction

Pinnatoxins (PnTxs) are lipophilic cyclic imine toxins that can accumulate in edible marine shellfish. They have been initially isolated from the shellfish *Pinna attenuata* collected in China [1]. Pinnatoxin A (PnTx-A) has been firstly characterized after its isolation from the bivalve *Pinna muricata* collected in Okinawa (Japan), similarly to its structural analogues PnTx-B, -C, and -D [2,3,4]. PnTx-E, -F, and -G have been subsequently isolated from oysters (*Crassostrea gigas*) collected in South Australia and New Zealand [5], whereas PnTx-H has been purified from cultured *Vulcanodinium rugosum* collected in South China Sea [6]. During the last years, these toxins, including their fatty acid ester derivatives, have also been detected in microalgae and/or other shellfish species from different geographical areas [7,8,9,10,11,12,13,14,15,16,17,18,19]. Moreover, strains of dinoflagellates producing PnTx-E and -F in New Zealand [7], have been shown to be identical to those producing PnTx-E, -F, and -G in Australia [8] or PnTx-G in Japan [9]. The isolates from New Zealand, Australia and Japan have been shown as morphologically identical to a Mediterranean dinoflagellate strain recognized as *Vulcanodinium rugosum* [20], a new species identified as pinnatoxins-producing organism [21].

Despite their accumulation in edible shellfish, pinnatoxins have not been associated to seafood poisonings in humans and, consequently, they are not currently regulated [17]. Nevertheless, because of their toxic potential, the European Food Safety Authority (EFSA) recommended more information on the oral toxicity of these compounds for a conclusive risk assessment [22]. In fact, pinnatoxins have been denoted as fast-acting and highly toxic compounds after intraperitoneal injection (i.p.) in mouse bioassays, being able to induce a rapid onset of neuromuscular effects leading to death. The toxicity is ascribed mainly to a specific interaction and blockage of muscle- and neuronal-type nicotinic acetylcholine receptors, known to be involved in synaptic transmissions at central and peripheral nervous system as well as at skeletal neuromuscular junction level. In particular, PnTXs bind muscle nicotinic acetylcholine receptors more potently than neuronal ones [23,24,25,26,27,28,29,30]. Intraperitoneal injection of these toxins in mice has been shown to induce skeletal muscle paralysis, respiratory depression, and lethal effects within less than one hour. The median lethal dose (LD_50_) was lower than 100 µg kg^−1^ with the following order of potency: PnTx-F > PnTx-G > PnTx-E > PnTx-H [5,6,31]. Pinnatoxins are characterized also by a high acute oral toxicity, which order rank after gavage administration in mice is PnTx-F > PnTx-G > PnTx-H > PnTx-E. The most toxic compounds (PnTx-F, G and H) have also been shown to present small differences in potency between oral and i.p. administration: their oral toxicity was only 1.6 to 9.3-fold lower than that after i.p. injection, whereas PnTx-E was at least 49-fold less toxic orally than after i.p. injection [6,31,32]. 

However, the few acute oral toxicity studies on pinnatoxins evaluated only their lethality and clinical signs, using non-fasted or 16 h-fasted mice and different administration routes. In particular, while no information on the experimental conditions are reported for PnTx-H toxicity assessment (LD_50_ = 163 µg kg^−1^), the LD_50_ of PnTx-E, -F and -G has been determined in female Swiss mice under different conditions: (1) in non-fasted mice administered by gavage with PnTx-E, -F or -G (LD_50_ = 2800, 25 and 150 µg kg^−1^, respectively); (2) in non-fasted mice that voluntary ingested PnTx-F or PnTx-G mixed with cream cheese (LD_50_ = 50 and 400 µg kg^−1^, respectively) or PnTx-F mixed with peanut butter (LD_50_ = 50 µg kg^−1^); (3) in 16 h-fasted mice administered with PnTx-F by gavage (LD_50_ = 29.9 µg kg^−1^), or by voluntary ingestion through mouse food (LD_50_ = 50 µg kg^−1^), through cream cheese (LD_50_ = 77 µg kg^−1^) or through a mixture of peanut butter, casein, and sucrose (LD_50_ = 50 µg kg^−1^) [6,31]. These data indicate that food/gastric content affect the oral toxicity of pinnatoxins in mice, probably interfering with their gastrointestinal absorption and bioavailability. This is suggested also by previous studies in mice showing that oral exposure to dinoflagellates extracts containing pinnatoxins by voluntary feeding induced lower toxicity than the same extracts administered by gavage [7,8]. Thus, the hazard of pinnatoxins after oral exposure should be firstly characterized in conditions minimizing the variability in investigatory parameters and discomfort of animals, such as in 3 h-fasted mice administered with the toxins by gavage to reduce the influence of food on their toxicity, as recommended by the guidelines of the Organization for Economic Co-operation and Development (OECD) for the testing of chemicals [33]. 

On the basis of this consideration, we challenged mice with PnTx-G through the oral route and we recorded the clinical signs, lethality, morphological changes in the main organs and tissues, and quantified selected blood chemistry parameters.

## 2. Results

### 2.1. Lethality and Clinical Signs

PnTx-G has been administered by gavage to groups of three mice at the doses of 8, 20, 50, and 120 μg kg^−1^, and we recorded the clinical signs and lethality up to 24 h. Subsequently, additional groups of 5 mice have been administered with PnTx-G at 220, 300, 370, and 450 μg kg^−1^ to calculate the toxin median lethal dose (LD_50_), based on 24 h lethality data.

At the dose of 220 µg kg^−1^ and above, PnTx-G administration provoked a rapid onset of clinical signs (prostration, tremors, jumping, followed by abdominal breathing, paralysis of the hind limbs and cyanosis), which led to the death of mice within less than 30 min. In particular, the lowest lethal dose (220 µg kg^−1^) provoked the death of 3/5 mice, while 370 μg kg^−1^ was lethal for 5/5 mice (Table 1). These results are presented in Figure 1 as percentage of mice mortality *versus* the administered toxin doses. Based on lethality data, the oral LD_50_ of PnTx-G was calculated at 208 μg kg^−1^ (95% confidence limits: 155–281 µg kg^−1^).

### 2.2. Blood Chemistry

Blood chemistry analyses did not show significant differences in the serum levels of aspartate aminotransferase (AST), alanine aminotransferase (ALT), glutamate dehydrogenase (GLDH), creatine phosphokinase (CPK), creatinine, and electrolytes (Na^+^, K^+^, Ca^2+^, Cl^−^, and inorganic phosphorous, P_i_) between control mice and those treated with PnTx-G (Table 2). 

### 2.3. Necropsy

No macroscopic alteration was observed in the organs of mice administered with any dose of PnTx-G (data not shown).

### 2.4. Histological Analysis

Histological analysis of the main organs and tissues showed tissue changes only in the small intestine of mice administered with PnTx-G at the dose of 300 μg kg^−1^ and above. In particular, moderate focal degeneration of the epithelial cells at the tips of the villi with villous atrophy was observed in all the animals administered with these toxin doses (Figure 2). 

## 3. Discussion

In 2011, *Vulcanodinium rugosum* has been identified as a harmful dinoflagellate species in French Mediterranean Ingril Lagoon [20]. In this geographical site, it has been shown to produce mainly PnTx-G, a cyclic imine responsible for the recurrent atypical shellfish toxicity at the mouse bioassay [13]. Similar to other analogues, PnTx-G is a fast-acting and highly toxic compound in mice, inducing a rapid onset of neuromuscular lethal effects after acute intraperitoneal or oral administration [5,6,31]. Despite pinnatoxins not being regulated as seafood contaminants [17], the European Food Safety Authority as well as the Food and Agriculture Organization of the United Nations addressed the need of new data for a conclusive risk assessment of these toxins, including information on their oral toxicity [22,34]. The few acute oral toxicity studies carried out on PnTx-E, -F, -G, and -H evaluated only their lethal potency and the clinical signs in mice [6,31]. PnTx-G administered by gavage to non-fasted mice resulted about 2.7-fold more toxic after gavage administration to non-fasted mice than by ingestion through voluntary intake of cream cheese [31]. Thus, to improve hazard characterization of this toxin, 3 h-fasted mice were administered by gavage with PnTx-G, in accordance with the OECD guidelines [33]. 

At the dose of 220 μg kg^−1^ and above, PnTx-G induced lethal effects within 30 minutes after its administration. Before death, mice showed piloerection, prostration, hypothermia, abdominal breathing, paralysis of the hind limbs, and cyanosis, consistent with nicotinic receptors blocking action by PnTx-G [23,24,25,26,27,28,29,30]. No signs of toxicity were recorded in survived mice. The median lethal dose (LD_50_) of PnTx-G was 208 μg kg^−1^ (95% confidence limits: 155–281 μg kg^−1^), comparable to that previously recorded in non-fasted mice by gavage administration (LD_50_ = 150 μg kg^−1^; 95% confidence limits: 105–199 μg kg^−1^), but significantly lower than that determined after voluntary intake through cream cheese mixture (LD_50_ = 400 μg kg^−1^; 95% confidence limits: 380–470 μg kg^−1^) [31]. These data indicate that the length of mice fasting does not influence the acute oral toxicity of PnTx-G after gavage administration, contrary to cream cheese that significantly reduces its toxicity. Mice administered with PnTx-G did not show any morphological alteration visible by necropsy, whereas light microscopy showed structural changes in the small intestine (moderate focal epithelial cells degeneration with villous atrophy) of mice administered with the dose of 300 μg kg^−1^ and above. It cannot be excluded that this effect could be related to an interaction of PnTx-G with nicotinic acetylcholine receptors at local or systemic level, but no information of the toxin kinetics supporting this hypothesis is available. Anyway, the focal morphological alterations should be considered for their potential impact on the barrier and permeability function of the intestinal epithelium. No significant changes in serum markers of liver, kidney, and/or muscle damage (AST, ALT, GLDH, creatinine, CPK) or of selected ions (Na^+^, K^+^, Ca^2+^, Cl^−^, P_i_) as indices of electrolytes homeostasis have been recorded. The absence of significant morphological tissues alterations in the main organs could be related to a fast functional damage induced by the toxin, such as neuromuscular block consequent to the antagonistic effect on nicotinic acetylcholine receptors [23,24,25,26,27,28,29,30], rather than a structural tissue damage. On the other hand, the nicotinic receptors antagonism might be at the basis of the main neuromuscular signs of mice before death (prostration, respiratory depression and hind limbs paralysis).

In conclusion, according to the results of this study, a provisional NOEL (no observed effect level) of PnTx-G after acute oral administration in mice can be estimated at 120 μg kg^−1^. Nevertheless, the LD_50_ of 208 μg kg^−1^ indicates a quite steep dose-toxicity relationship. Thus, considering the high toxic potential of PnTx-G and its frequent detection in edible shellfish from Mediterranean Sea, further toxicity studies after repeated oral exposure are necessary for the risk assessment of this compound as seafood contaminant.

## 4. Materials and Methods 

### 4.1. Toxin, Reagents, and Other Chemicals

PnTx-G has been isolated from *Vulcanodinium rugosum*. An aliquot of the pure material has been characterized by nuclear magnetic resonance (NMR), gravimetry, and liquid chromatography mass spectrometry (LC-MS) and tandem mass spectrometry. No significant impurities were detected by NMR and LC-MS analysis. The purified toxin was dissolved in methanol containing 0.01% acetic acid and stored in flame-sealed 2.5 mL amber glass ampoules at −20 °C. Stability of the toxin was tested in an accelerated stability study at temperatures up to 40 °C for 2 weeks and no degradation was detected over this period. 

Tiletamine/zolazepam (Zoletil^®^) and xylazine (Virbaxyl^®^) were purchased from Virbac (Milan, Italy). Other chemicals were obtained from Sigma Aldrich (Milan, Italy).

### 4.2. Animals and Experimental Conditions

Female CD-1 mice (18–20 g body weight, 4 weeks old), bedding and feed were purchased from Harlan Laboratories (S. Pietro al Natisone, UD, Italy). Animals were acclimatized for one week before treatment in the animal room set to maintain controlled temperature (21 ± 1 °C) and humidity (60–70%), in the presence of a fixed artificial light cycle (7.00 a.m.–7.00 p.m.). Mice were caged in groups of 3 or 5 animals, using dust free poplar chips for bedding and fed a standard diet for rodents. Water was provided *ad libitum* during all phases of the study. Experiments were carried out at the University of Trieste in conformity with Italian D.L. n. 116 of 27 January 1992 and associated guidelines in the European Communities Council Directive of 24 November 1986 (86/609 ECC) concerning animal welfare and appendix A of the European Convention ETS 123.

### 4.3. Experimental Design

Three-hours fasted mice were weighed immediately before treatment. PnTx-G, dissolved in phosphate buffered saline (PBS) pH 7.0 containing 1.8% ethanol (*v/v*), was administered by gavage to groups of 3 or 5 mice (administered volume: 10 mL kg^−1^, adjusted on the basis of mouse weight). Control animals received the vehicle alone at the same volume. Feed was returned within 2 h after dosing and, during the observation period, it was offered *ad libitum*. 

After dosing, mice were continuously observed for 24 h, recording mortality and signs of toxicity. Then, mice were weighed and anesthetized by intraperitoneal injection of tiletamine/zolazepam (Zoletil^®^; 20 mg kg^−1^) and xylazine (Virbaxyl^®^; 5 mg kg^−1^). Animals were then exsanguinated for blood chemistry analyses and submitted to gross necropsy. The main organs and tissues were removed and fixed in neutral buffered 10% formalin for the histological analysis. Similarly, mice died during the observation period were immediately weighed and necropsied; blood was collected for hematochemical analyses, and the main organs and tissues were removed and fixed for the histological analysis. The experimental study has been approved by the Animal Welfare Body of the University of Trieste (review board: Piero Paolo Battaglini, Tullio Giraldi, Paola Lorenzon, Paola Zarattini, Davide Barbetta) and the Italian Ministry of Health (decree n. 112/2013-B of 14th May 2013).

### 4.4. Blood Chemistry

Blood was allowed to clot for 15 min at room temperature, and centrifuged at 2000× *g* for 10 min at 4 °C. Serum was collected and stored at −80 °C until the analyses were carried out. The serum levels of aspartate aminotransferase, alanine aminotransferase, glutamate dehydrogenase, creatine phosphokinase, creatinine, and electrolytes (Na^+^, K^+^, Ca^2+^, Cl^−^ and inorganic phosphate, P_i_) were measured using an automatized chemistry analyzer (AU400 Olympus with Beckman Coulter reagents; Olympus America Inc., Melville, NY, USA).

### 4.5. Histological Analysis

Heart, liver, lungs, kidneys, spleen, stomach, duodenum, jejunum, colon, rectum, pancreas, thymus, cerebrum, cerebellum, spinal cord, uterus, ovaries, and skeletal muscle (soleus), fixed in 10% neutral buffered formalin, were dehydrated, embedded in paraffin and cut in sections of 5 µm. Sections were stained with hematoxylin-eosin. Blind histopathological examination was carried out. Pictures were obtained with a Nikon eclipse i50 microscope equipped with a DS-Vi1 digital camera and NIS-Elements Microscope Imaging Software.

### 4.6. Statistical Analysis and Determination of LD_50_

Data are expressed as mean ± standard error (S.E.). Significant differences between control and experimental groups were calculated by one-way analysis of variance, followed by the Dunnett’s post-test for multiple comparisons of unpaired data, accepting *p* < 0.05 as significant. LD_50_, based on 24-h mortality data, was calculated according to the Finney method [35] at a 95% confidence level, using Elsevier-Biosoft software (Cambridge, UK).

## Figures and Tables

**Figure 1 toxins-12-00087-f001:**
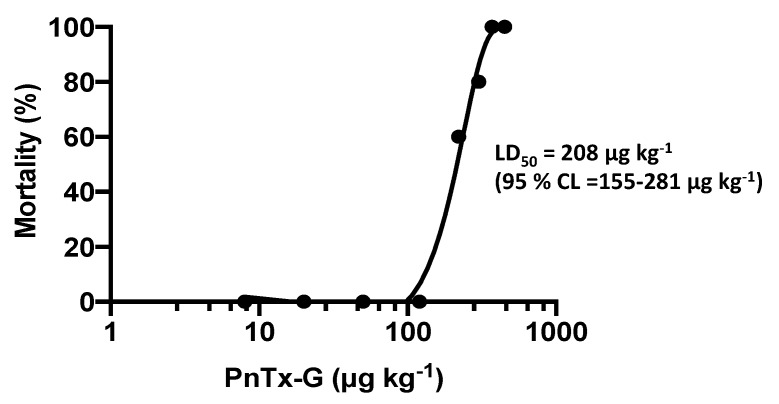
Dose-response mortality curve of PnTx-G after acute oral administration in mice. Percentage lethality is plotted against the administered doses of PnTx-G.

**Figure 2 toxins-12-00087-f002:**
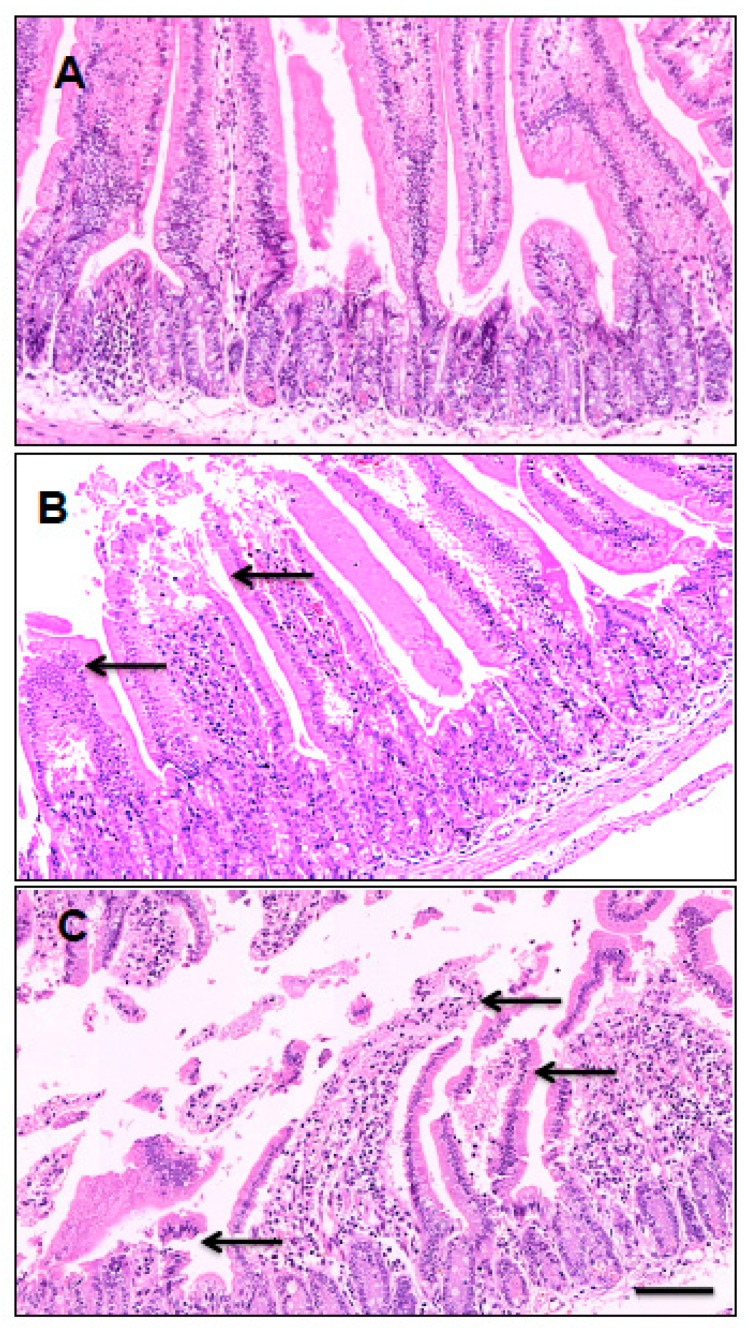
Light micrographs of cross sections of the small intestine from control mouse (**A**), and PnTx-G-treated mice at 300 μg kg^−1^ (**B**) or 450 μg kg^−1^ (**C**). Haematoxylin-eosin stain; objective magnification: 10 ×: scale bar: 20 μm.

**Table 1 toxins-12-00087-t001:** Lethality and signs of toxicity of mice after acute oral administration of PnTx-G.

Group of Treatment	Dose (µg kg^−1^)	Lethality	Survival Times (min)	Signs of Toxicity
Controls	-	0/8	-	-
PnTx-G	8	0/3	-	-
	20	0/3	-	-
	50	0/3	-	-
	120	0/3	-	-
	220	3/5	20-22-22	Prostration, tremors, jumping, abdominal breathing, paralysis of the hind limbs, cyanosis
	300	4/5	12-13-17-23	Prostration, tremors, jumping, abdominal breathing, paralysis of the hind limbs, cyanosis
	370	5/5	13-15-16-17-18	Prostration, tremors, jumping, abdominal breathing, paralysis of the hind limbs, cyanosis
	450	5/5	12-12-15-16-29	Prostration, tremors, jumping, abdominal breathing, paralysis of the hind limbs, cyanosis

**Table 2 toxins-12-00087-t002:** Blood chemistry parameters of mice after acute oral administration of PnTx-G.

Parameter	Controls (*n* = 8)	PnTx-G 8 μg kg^−1^ (*n* = 3)	PnTx-G 20 μg kg^−1^ (*n* = 3)	PnTx-G 50 μg kg^−1^ (*n* = 3)	PnTx-G 120 μg kg^−1^ (*n* = 3)	PnTx-G 220 μg kg^−1^ (*n* = 3)	PnTx-G 300 μg kg^−1^ (*n* = 5)	PnTx-G 370 μg kg^−1^ (*n* = 3)	PnTx-G 450 μg kg^−1^ (*n* = 4)
ALT (IU/L)	58.1 ± 11.0	34.7 ± 9.6 *(−40%)*	37.7 ± 5.4 *(−35%)*	34.0 ± 6.1 *(−41%)*	29.3 ± 4.7 *(−50%)*	65.5 ± 14.9 *(13%)*	63.3 ± 14.0 *(9%)*	43.7 ± 4.7 *(−25%)*	63.3 ± 13.6 *(9%)*
AST (IU/L)	77.8 ± 8.4	80.7 ± 29.2 *(4%)*	78.7 ± 11.7 *(1%)*	88.0 ± 11.8 *(13%)*	93.3 ± 28.8 *(20%)*	95.3 ± 16.0 *(22%)*	97.3 ± 14.6 *(25%)*	96.3 ± 8.6 *(24%)*	104.3 ± 23.4 *(34%)*
GLDH (IU/L)	46.8 ± 12.8	40.5 ± 3.8 *(−13%)*	33.4 ± 2.7 *(−27%)*	31.1 ± 0.3 *(−34%)*	35.0 ± 3.0 *(−25%)*	52.8 ± 11.8 *(13%)*	25.2 ± 6.0 *(−46%)*	35.2 ± 3.0 *(−25%)*	48.9 ± 14.1 *(5%)*
CPK (IU/L)	1158.2 ± 212.8	597.0 ± 199.8 *(−48%)*	1365.3 ± 639.1 *(18%)*	670.7 ± 114.2 *(−42%)*	1381.0 ± 54.0 *(19%)*	1126.3 ± 290.6 *(−3%)*	1245.0 ± 330.6 *(7%)*	1176.0 ± 100.7 *(2%)*	1131.8 ± 138.8 *(−2%)*
Creatinine (mg/dL)	28.1 ± 1.3	27.7 ± 0.9 *(−1%)*	26.0 ± 1.5 *(−7%)*	25.7 ± 1.9 *(−9%)*	25.7 ± 1.8 *(−9%)*	40.6 ± 12.9 *(47%)*	24.7 ± 1.2 *(−12%)*	27.7 ± 2.0 *(1%)*	33.3 ± 3.4 *(19%)*
Ca^2+^ (mM)	2.9 ± 0.1	2.5 ± 0.1 *(−14%)*	2.6 ± 0.2 *(−10%)*	2.6 ± 0.1 *(−10%)*	2.5 ± 0.1 *(−14%)*	3.1 ± 0.2 *(7%)*	2.8 ± 0.2 *(−3%)*	3.2 ± 0.3 *(10%)*	3.2 ± 0.3 *(10%)*
Na^+^ (mM)	136.3 ± 1.5	137.5 ± 0.8 *(1%)*	133.8 ± 1.5 *(−2%)*	134.2 ± 1.6 *(−2%)*	136.3 ± 2.3 *(0%)*	136.3 ± 5.9 *(0%)*	124.1 ± 9.4 *(−9%)*	138.2 ± 0.6 *(1%)*	133.9 ± 3.9 *(−3%)*
K^+^ (mM)	9.3 ± 0.8	6.6 ± 0.3 *(−29%)*	8.4 ± 0.9 *(−10%)*	7.6 ± 0.2 *(−18%)*	7.3 ± 0.8 *(−47%)*	10.4 ± 1.8 *(12%)*	7.2 ± 1.3 *(−23%)*	7.1 ± 3.7 *(8%)*	10.3 ± 0.4 *(−11%)*
Cl^−^ (mM)	103.2 ± 1.0	103.6 ± 1.8 *(1%)*	101.9 ± 1.0 *(−2%)*	101.0 ± 2.2 *(−2%)*	103.7 ± 2.3 *(1%)*	104.8 ± 2.6 *(2%)*	102.6 ± 4.3 *(−1%)*	107.4 ± 1.1 *(4%)*	102.9 ± 2.8 *(−1%)*
P_i_ (mM)	3.9 ± 0.2	3.2 ± 0.2 *(−18%)*	3.6 ± 0.2 *(−8%)*	3.4 ± 0.2 *(−13%)*	3.2 ± 0.3 *(−18%)*	4.0 ± 0.4 *(3%)*	4.3 ± 0.3 *(10%)*	4.5 ± 0.4 *(15%)*	4.5 ± 0.8 *(15%)*

Data are the mean ± standard error (S.E.); in brackets: % differences with respect to controls; values not significantly different from those of controls, at the analysis of variance.

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
