# Peer review of "Acute Oral Toxicity of Pinnatoxin G in Mice"

_toxins, 2020, doi:10.3390/toxins12020087_

Round 1

Reviewer 1 Report

This article provides information on the toxicity of PnTX-G, a toxin for which not much toxicity information has been gathered to date.  While the article satisfies a gap in knowledge, it may only be of interest to individuals in areas in which PnTX-G has been detected.

The majority errors detected in the manuscript concerned grammatical mistakes. 

In the abstract, line 8, is this a sentence regarding the "potential" harmful mechanism of toxicity?  The sentence prior and after gives the context suggestion that it is potential and not definitive.  Rewording those sentences to make the intention clearer is needed.

Abstract, line 8, remove "Anyway".  Too many modifying phrases throughout the manuscript and this particular one is too conversational for a formal manuscript.  

Introduction, line 32, place also before been and eliminate the also before in microalgae

Introduction, line 35, remove then.  Not needed for the sentence.

Introduction, line 44,"blockage" instead of "block"

Discussion, line 115, remove "-ly" at end of similarly

Discussion, line 118, remove anyway.

Author Response

Comment. In the abstract, line 8, is this a sentence regarding the "potential" harmful mechanism of toxicity? The sentence prior and after gives the context suggestion that it is potential and not definitive. Rewording those sentences to make the intention clearer is needed. Abstract, line 8, remove "Anyway". Too many modifying phrases throughout the manuscript and this particular one is too conversational for a formal manuscript.  

Answer. We thank the reviewer for his/her suggestion to make the intention clearer. By he sentence at line 8, we wanted to indicate the actual harmful mechanism of toxicity. According to the reviewer suggestion, we re-phrased the sentences as follows.

“As other pinnatoxins, to date no human poisonings ascribed to consumption of PnTx-G contaminated seafood have been reported, despite its potent antagonism at nicotinic acetylcholine receptors and its high and fast-acting toxicity after intraperitoneal or oral administration in mice”.

Comments. Introduction, line 32, place also before been and eliminate the also before in microalgae.

Introduction, line 35, remove then.  Not needed for the sentence.

Introduction, line 44,"blockage" instead of "block".

Discussion, line 115, remove "-ly" at end of similarly

Discussion, line 118, remove anyway.

Answer. We thank the reviewer for all these suggestions, which have been considered to modify the relevant sentences.

Reviewer 2 Report

toxins-689661

The article entitled "Acute oral toxicity of pinnatoxin G in mice” is a toxicology study on mice, to characterize the toxicity of pinnatoxin-G, a dinoflagellate neurotoxin which can contaminate the food chain via different types of seafood.

The authors first carried out oral administration of increasing doses of PnTx-G to estimate the LD50, and the symptoms - mainly neurotoxic - associated. These results largely confirm those which have been published previously (doi: 10.1016/j.toxicon.2012.07.002). The most interesting in this study is the estimation of the lethality and the associated symptoms, which provide a NOEL. The data show that for all lethal doses (220 to 450 µg/kg), death is preceded by neurotoxic symptoms. For all non-lethal doses (8 to 120 µg/kg), no signs of toxicity were observed. How can you explain that there is no sublethal dose for observing signs of intoxication ? Can we imagine that this would have been the case for a dose between 120 and 220 µg/kg ?

In addition, a histological analysis of the main organs as well as biochemical blood analyzes were carried out. The biochemical blood analyzes did not show any difference compared to the control group. This raises questions because we note for certain parameters variations of about 50%, in particular for alanine aminotransferase (-40, -35, - 41, -50% for the lowest doses of PnTx-G), without any significant difference.

Small tissue changes were observed in the small intestine of mice exposed to lethal doses of PnTX-G (moderate mucosal degeneration, villous atrophy). The authors indicate that "only slight changes in the small intestine" are observed. When comparing Figures 1A and 1C, these differences seem to be quite large. The authors should comment better on these data, giving more details, especially since no positive control is carried out (with for example an agent that would induce major deleterious effects on intestinal integrity). Could this effect be due to the interaction of PnTx-G with nicotinic receptors locally? Does PnTx-G fully cross the intestinal barrier? Are data available on the toxicokinetics of PnTx-G? These issues are not discussed.

L46: in references, please insert doi: 10.3390/md17070425

L46: It should be added that “PnTXs bind muscle nicotinic ACh rceptors more potently than neuronal ones” (reviewed in [27]). L.123 to 128 of the discussion are very repetitive with L.58 to 65 of the introduction.

L205: 4.6. Statistical analysis and determination of LD50. Please give more details on the determination of the LD50 (software used, non linear regression fit etc.). Please insert under Table 1 or as a supplementary figure the LD50 curve (as in doi: 10.3390/toxins9030075).

Minor

Pinnatoxin is alternatively written “PnTX” or “pinnatoxin”. Harmonize its writing using PnTx.

L56: and different administration routes.

L74: On the basis of this consideration, we challenged mice with PnTx-G through the oral route.

L75-76: …, and we recorded clinical signs… and quantified selected blood…

L81: We recorded clinical signs and lethality up to 24 h.

L106: “degeneration of the small intestinal mucosa with villous atrophy”. Please add arrows on the micrographs for more clarity.

L109-110: Magnification is provided but insert a scale bar on the microcraphs.

L186: exsanguinated for blood chemistry analyses

Author Response

Comment. The authors first carried out oral administration of increasing doses of PnTx-G to estimate the LD50, and the symptoms - mainly neurotoxic - associated. These results largely confirm those which have been published previously (doi: 10.1016/j.toxicon.2012.07.002). The most interesting in this study is the estimation of the lethality and the associated symptoms, which provide a NOEL. The data show that for all lethal doses (220 to 450 µg/kg), death is preceded by neurotoxic symptoms. For all non-lethal doses (8 to 120 µg/kg), no signs of toxicity were observed. How can you explain that there is no sublethal dose for observing signs of intoxication? Can we imagine that this would have been the case for a dose between 120 and 220 µg/kg?

Answer. We cannot exclude the possibility of sublethal doses of PnTx-G within the range of 120-220 µg/kg that could induce signs of toxicity in mice. However, we recorded a steep dose-toxicity relationship, with an estimated NOEL of 120 mg/kg and a calculated LD50 of 208 mg/kg. Due to ethical and policy issues related to the use of animals, we cannot test further toxin doses within this range to verify this possibility.

Comment. In addition, a histological analysis of the main organs as well as biochemical blood analyzes were carried out. The biochemical blood analyzes did not show any difference compared to the control group. This raises questions because we note for certain parameters variations of about 50%, in particular for alanine aminotransferase (-40, -35, - 41, -50% for the lowest doses of PnTx-G), without any significant difference.

Answer. We agree with the reviewer, however, due to the variability of the values of these parameters within each experimental group, also due to the limited number of mice for each experimental group, these differences are not significant as compared to control mice. Anyway, the measured values of these parameters remained within the physiological range in mice (Suckow et al. – The Laboratory Mouse. CRC Press, Boca Raton 2001; Serfilippi et al.- Contemporary Topics in Laboratory Animal Science/American Association for Laboratory Animal Science, 42, 46-52). Furthermore, while an increased serum level of alanine aminotransferase is usually associated with a liver toxicity, we recorded a reduction in the mean serum level of this enzyme.

Comment. Small tissue changes were observed in the small intestine of mice exposed to lethal doses of PnTX-G (moderate mucosal degeneration, villous atrophy). The authors indicate that "only slight changes in the small intestine" are observed. When comparing Figures 1A and 1C, these differences seem to be quite large. The authors should comment better on these data, giving more details, especially since no positive control is carried out (with for example an agent that would induce major deleterious effects on intestinal integrity). Could this effect be due to the interaction of PnTx-G with nicotinic receptors locally? Does PnTx-G fully cross the intestinal barrier? Are data available on the toxicokinetics of PnTx-G? These issues are not discussed.

Answer. We thank the reviewer for this observation. Actually, we recorded moderate mucosal degeneration and villous atrophy in the small intestine; the wording “only slight changes” remained erroneously in the text. As suggested, we provided more information on these tissue changes in the Results section and comments in Discussion as reported below.

Considering the possible relation between the morphological changes at the small intestine and the interaction of PnTx-G with nicotinic receptors, we cannot exclude this possibility, but we do not have information to support this hypothesis, including toxicokinetic data. This aspect has also been commented in Discussion as reported below.

Results: “Histological analysis of the main organs and tissues showed tissue changes only in the small intestine of mice administered with PnTx-G at the dose of 300 μg kg-1 and above. In particular, moderate focal degeneration of the epithelial cells at the tips of the villi with villous atrophy was observed in all the animals administered with these toxin doses (Figure 2)”.

Discussion: These records were further commented in Discussion section as follows “Mice administered with PnTx-G did not show any morphological alteration visible by necropsy, whereas light microscopy showed structural changes in the small intestine (moderate focal epithelial cells degeneration with villous atrophy) of mice administered with the dose of 300 μg kg-1 and above. It cannot be excluded that this effect could be related to an interaction of PnTx-G with nicotinic acetylcholine receptors at local or systemic level, but no information on the toxin kinetics supporting this hypothesis is available, to the best of our knowledge. Anyway, the focal morphological alterations should be considered for their potential impact on the barrier and permeability function of the intestinal epithelium”.

Comment. L46: in references, please insert doi: 10.3390/md17070425.

Answer. According to the reviewer suggestion, we added the citation in the text and the reference in the list.

Comment. L46: It should be added that “PnTXs bind muscle nicotinic ACh receptors more potently than neuronal ones” (reviewed in [27]).

Answer. We thank the reviewer for this clarification that has been added in the text.

Comment. L.123 to 128 of the discussion are very repetitive with L.58 to 65 of the introduction.

Answer. Accordingly to the review suggestion, we modified this part of Discussion as follows. “PnTx-G administered by gavage to non-fasted mice resulted about 2.7-fold more toxic than its ingestion by voluntary intake through cream cheese [31]. Thus, to improve the hazard characterization of this toxin, 3 h-fasted mice were administered by gavage with PnTx-G, in accordance with the OECD guidelines [33”].

Comment. L205: 4.6. Statistical analysis and determination of LD50. Please give more details on the determination of the LD50 (software used, non linear regression fit etc.).

Answer. According to the reviewer suggestion, we reported the software (Elsevier-Biosoft software; Cambridge, UK) used to determine LD50.

Comment. Please insert under Table 1 or as a supplementary figure the LD50 curve (as in doi: 10.3390/toxins9030075).

Answer. According to the reviewer suggestion, the dose-response mortality curve with LD50 has been inserted under Table 1.

Comment. Pinnatoxin is alternatively written “PnTX” or “pinnatoxin”. Harmonize its writing using PnTx.

Answer. We thank the reviewer observation and, accordingly, we harmonized the abbreviation of pinnatoxin G using PnTx-G.

Comments. L56: and different administration routes.

L74: On the basis of this consideration, we challenged mice with PnTx-G through the oral route.

L75-76: …, and we recorded clinical signs… and quantified selected blood…

L81: We recorded clinical signs and lethality up to 24 h.

Answer. According to the reviewer’s suggestion, we changed the relevant sentences.

Comment. L106: “degeneration of the small intestinal mucosa with villous atrophy”. Please add arrows on the micrographs for more clarity. L109-110: Magnification is provided but insert a scale bar on the microcraphs.

Answer. According to the reviewer’s suggestions, we added arrows and a scale bar on the micrographs and the relevant description in the figure legend.

Comment. L186: exsanguinated for blood chemistry analyses

Answer. We thank the reviewer for the suggestion and accordingly we modified the sentence.

Reviewer 3 Report

The article deals with the estimation of the oral toxicity of PnTx G. Only some aspects are new but even the information on aspects that have already been studied is relevant for stablishing a solid regulation of this group of toxins.

The article is, en general, well written and understable, but the discussion section needs to be improved. It starts with redundant information of the introduction (it is a secon introduction) and the results of the histological alterations are nearly not discussed. Some comparisons with the alteration induced by other PnTXs and/or cyclic imines should be included.

Probably, a reference with the discovery of pinnatoxins by Zheng et al in 1990 should be included in the introduction.

"in vivo" should be removed from "in vivo oral"  (several times along the text) because there is no posibility of "in vitro" oral exposure.

L40 reword the sentence because a need cannot be recommended.

L67-68. This sentence is unclear. Reword it.

L86 "the lower" --> the lowest

Author Response

Comment. The article is, en general, well written and understable, but the discussion section needs to be improved. It starts with redundant information of the introduction (it is a second introduction) and the results of the histological alterations are nearly not discussed. Some comparisons with the alteration induced by other PnTXs and/or cyclic imines should be included.

Answer. As suggested also by Reviewer 2, we modified the initial part of the Discussion and discussed the histological findings, as reported below. However, due to the lack of literature data, we did not compare the alterations recorded in this study with those induced by the oral administration of other PnTXs or cyclic imines.

“In 2011, Vulcanodinium rugosum has been identified as a harmful dinoflagellate species in French Mediterranean Ingril Lagoon [20]. In this geographical site, it has been shown to produce mainly PnTx-G, a cyclic imine responsible for the recurrent atypical episodes of shellfish toxicity at the mouse bioassay [13]. Similar to other analogues, PnTx-G is a fast-acting and highly toxic compound in mice, inducing a rapid onset of neuromuscular lethal effects after acute intraperitoneal or oral administration [5, 6, 31]. Despite pinnatoxins not being regulated as seafood contaminants [17], the European Food Safety Authority as well as the Food and Agriculture Organization of the United Nations addressed the need of new data for a conclusive risk assessment of these toxins, including information on their oral toxicity [22, 34]. The few acute oral toxicity studies carried out on PnTx-E, -F, -G and -H evaluated only their lethal potency and the clinical signs in mice [6, 31]. PnTx-G administered by gavage to non-fasted mice resulted about 2.7-fold more toxic after gavage administration to non-fasted mice than by ingestion through voluntary intake of cream cheese [31]. Thus, to improve hazard characterization of this toxin, 3 h-fasted mice were administered by gavage with PnTx-G, in accordance with the OECD guidelines [33]”. …..

“Mice administered with PnTx-G did not show any morphological alteration visible by necropsy, whereas light microscopy showed structural changes in the small intestine (moderate focal epithelial cells degeneration with villous atrophy) of mice administered with the dose of 300 mg kg-1 and above. It cannot be excluded that this effect could be related to an interaction of PnTx-G with nicotinic acetylcholine receptors at local or systemic level, but no information of the toxin kinetics supporting this hypothesis is available, to the best of our knowledge. Anyway, the focal morphological alterations should be considered for their potential impact on the barrier and permeability function of the intestinal epithelium”.

Comment. Probably, a reference with the discovery of pinnatoxins by Zheng et al in 1990 should be included in the introduction.

Answer. As properly suggested, the citation has been included in the Introduction section.

Comment. "in vivo" should be removed from "in vivo oral"  (several times along the text) because there is no possibility of "in vitro" oral exposure.

Answer. We agree with the reviewer and, accordingly we removed “in vivo” from “in vivo oral”.

Comments. L40 reword the sentence because a need cannot be recommended. L67-68. This sentence is unclear. Reword it. L86 "the lower" --> the lowest.

Answer. We thank the reviewer and, according to his/her suggestions, we reworded the sentences and changed “the lower” with “the lowest”.

Round 2

Reviewer 2 Report

The modifications suggested in Round 1 have been taken into account. It may be published in its present form.